# Epipyrone A, a Broad-Spectrum Antifungal Compound Produced by *Epicoccum nigrum* ICMP 19927

**DOI:** 10.3390/molecules25245997

**Published:** 2020-12-18

**Authors:** Alex J. Lee, Melissa M. Cadelis, Sang H. Kim, Simon Swift, Brent R. Copp, Silas G. Villas-Boas

**Affiliations:** 1School of Biological Sciences, University of Auckland, 3A Symonds Street, 1010 Auckland, New Zealand; alexleenz@hotmail.com (A.J.L.); sam.sangheon.kim@gmail.com (S.H.K.); 2School of Chemical Sciences, University of Auckland, 23 Symonds Street, 1010 Auckland, New Zealand; m.cadelis@auckland.ac.nz (M.M.C.); b.copp@auckland.ac.nz (B.R.C.); 3School of Medical Sciences, University of Auckland, 85 Park Road, Grafton, 1023 Auckland, New Zealand; s.swift@auckland.ac.nz

**Keywords:** polyenes, pigment, fungicide, antimicrobial, yeast, antibiotic, bioactives

## Abstract

We have isolated a filamentous fungus that actively secretes a pigmented exudate when growing on agar plates. The fungus was identified as being a strain of *Epicoccum nigrum*. The fungal exudate presented strong antifungal activity against both yeasts and filamentous fungi, and inhibited the germination of fungal spores. The chemical characterization of the exudate showed that the pigmented molecule presenting antifungal activity is the disalt of epipyrone A—a water-soluble polyene metabolite with a molecular mass of 612.29 and maximal UV–Vis absorbance at 428 nm. This antifungal compound showed excellent stability to different temperatures and neutral to alkaline pH.

## 1. Introduction

*Epicoccum* species are primarily saprophytic fungi from the family Didymellaceae. These fungi have been isolated from virtually all possible terrestrial and marine substrates worldwide [1,2,3]. Substantial intraspecific morphological and genetic diversity have been reported for this genus [2]. *Epicoccum nigrum* Link 1816 (synonym *E. purpurascens* Ehrenb. 1818) is well known for producing numerous secondary metabolites [4,5,6,7,8]. The antifungal activities of some of these compounds led to attempts to develop biological control products based on *E. nigrum* mycelia, spores and metabolites [9,10,11,12].

Several biological active metabolites produced by different *E. nigrum* strains have been characterized previously [5,13,14,15,16,17,18]. The most well-known is flavipin (3,4,5-trihydroxy-6-methyl-O-phthalaldehyde), which is produced by some *E. nigrum* strains and also by other fungi [13,14]. Flavipin presents strong antimicrobial activity against both bacteria and fungi [13,14]. Epicoccins [15], epicorazines [5] and epirodin [16] are produced by some *E. nigrum* strains, which also present moderate antibacterial activity. On the other hand, epicoccamides [8,17] and thiornicin [18] have been reported as presenting anti-tumour properties.

Many *Epicoccum* species secrete different pigments during growth [6,13,19]. Four carotenoids, including ß-carotene and γ-carotene, were detected as some of the pigmented metabolites produced by *Epicoccum nigrum* [19]. Epicocconone, a fluorescent-pigmented compound, is produced by some *Epicoccum* species and has been used as a protein-binder dye [6]. We have chemically characterized a yellow water-soluble pigment compound produced by a strain of *E. nigrum* isolated in New Zealand, which exhibits strong and broad antifungal activity. We demonstrate here that the pigmented molecule presenting antifungal activity is epipyrone A (disalt) (compound **1**, Figure 1), a molecule previously reported as the neutral diacid (compound **2**, Figure 1) with antiviral and telomerase inhibitory properties isolated from an unidentified *Epicoccum* species [3,20]. Here, however, we demonstrate that the disalt form of epipyrone A is a thermo-stable and inhibits both yeast and fungal mycelial growth, and the germination of fungal spores.

## 2. Results

### 2.1. Purification, Chemical Properties and Antifungal Activity

*E. nigrum* when cultivated on Czapek Yeast Agar (CYA) plates actively secretes a pigmented exudate (Figure 2A). When dissolved in water or polar solvents, the exudate develops into a dark orange to bright yellow solution depending on the concentration of exudate (Figure 2B). This pigmented solution showed antifungal activity against other filamentous fungi and yeasts (Figure 3). No antimicrobial activity against bacteria was however observed.

A chromatographic analysis of the methanol extract obtained from *E. nigrum* cultures indicated the presence of two major pigmented compounds (peak 1 and 2, Figure 4A) and at least two other less abundant yellow compounds (Figure 4A). The two major peaks were collected separately through semi-preparative HPLC, and then re-injected into the HPLC to confirm their purity. Interestingly, the same major peak 1 and a secondary peak 2 were observed when each peak was re-dissolved in analytical methanol and re-analysed (Figure 4B,C). When dissolved in Milli-Q H_2_O however, only the secondary peak 2 was observed from both fractions suggesting an interconvertible structural relationship between peak 1 and secondary peak 2 compounds, which probably depended on the ionization strength of the solvent used (Figure 4D,E). High-resolution mass spectrometry analysis of both peaks showed they were indeed molecular isomers with a nominal mass of 612.29 (found 613.2969 for MH+) and presented identical fragmentation patterns obtained from MS-MS analysis, suggesting therefore they were the same molecule or stereoisomers.

Both interconvertible yellow compounds (peak 1 and 2, Figure 4) were readily soluble in water and polar organic solvents such as methanol and ethanol. They were found to be unstable under acidic conditions changing from a bright yellow colour to a pale orange. However, when an acidic solution containing either compound was adjusted to pH 12, it recovered its bright yellow colour, suggesting a reversible pH-dependent structural rearrangement. The compounds remained relatively stable in the presence of light for over 10 days in aqueous solutions, with a slight degradation when dissolve in H_2_O at pH 7 (Table 1). The yellow compounds also appear to be resistant to high temperature up to 100 °C and microwave radiation (1000 W for 5 min) (Table 1).

To confirm the successful purification of the antifungal compound from the crude methanol extract of *E. nigrum*, large volume of crude methanol extract was subjected to column chromatography. High resolution mass spectrometric analysis permitted to estimate its molecular formula of C_34_H_44_O_10_ based on its sodiated peak at *m*/*z* 635.2825 [M + Na] (calcd 635.2830). The purified yellow compound (dissolved in water, and therefore equivalent to HPLC peak shown in Figure 4D,E) was then subjected to agar diffusion assays against a series of filamentous fungi and yeasts (Table 2). We observed a clear zone of inhibition against all yeasts and moulds tested (Table 2 and Figure 5), confirming its antifungal activity against a wide range of fungal species. The compound was also able to completely inhibit the spore germination of *B. cinera* during incubation in rich culture medium for 10 days (Figure 6). All three different concentrations of purified compound tested were able to completely inhibit spore germination of *B. cinera*, which indicates its minimal inhibition concentration (MIC) to be below 0.7 mg.mL^−1^. Nevertheless, when the biological activity assays described above were carried out at pH below 6, no antifungal activity was observed.

### 2.2. Chemical Structure Characterisation

Analysis of the ^1^H and ^13^C NMR data (Appendix A) showed striking similarity to those reported for the polyene orevactaene/epipyrone A [3]. The structure of orevactaene, a polyene originally reported to contain an α-pyrone fused with a galactopyranosyl ring [21], was later discovered to be identical to epipyrone A, a polyene with an unfused ring system, after total synthesis of the both natural products [22]. While the majority of ^1^H and ^13^C NMR chemical shifts observed for compound **1** (Figure 1) were similar to those reported for synthetic epipyrone A (compound **2**, Figure 1) [22], resonances associated with the pyrone ring were noticeably different (Table 3). To facilitate complete assignment of the carbon skeleton of the molecule, the fungus was cultured on minimal media supplemented with [U-^13^C]glucose [23] Purification of the disalt using RP-8 column chromatography again afforded 1 (Figure 1), this time highly enriched with ^13^C. Connectivity of the contiguous carbon backbone of compound **1** was determined by analysis of a magnitude mode ^13^C-^13^C COSY (Bruker pulse sequence cosydcqf) data set [24]. Tracing the carbon-carbon connectivity, derived from the neighbouring carbon atom correlations observed in the ^13^C-^13^C COSY NMR experiment and a ^1^H-^13^C HMBC NMR experiment (Appendix A), afforded unambiguous assignment the backbone of the molecule. While the majority of the ^13^C chemical shifts observed for compound **1** were similar to those reported for epipyrone A (Table 3) [3], notable differences (δ_compound **1**_ − δ_epipyrone A_ ≥ 7 ppm) were centred upon the 4-hydroxy-2-pyrone [C-3 (δΔ + 12.7), C-4 (δΔ + 9.3)] and carboxylic acid [C-27 (δΔ + 7.6)] moieties (Figure 1). In the case of the pyrone ring, the chemical shift observed for C-3 (δc 181.8) was suggestive of a (gamma) 4-pyrone resonance structure [25], while the difference in the carboxylic acid chemical shift led us to conclude it was present in compound **1** as the carboxylate anion. To assess this latter hypothesis, compound **1** was stirred in a solution of trifluoroacetic acid (10%) in methanol for 1 h, dried to give a dark orange oil and then re-examined by NMR. The ^1^H and ^13^C NMR data observed for the acid-treated compound were found to be identical to those previously reported for epipyrone A [3]. We thus concluded that compound **1** originally isolated from the methanol extract of *E. nigrum* and re-dissolved in water was a disalt (presumably sodium) of epipyrone A (Figure 1). What is interesting is that the salt form of the 4-hydroxy-2-pyrone moiety exists as a (gamma) 4-pyrone resonance structure, as evidenced by the chemical shift observed for C-3 (δc 181.8). Therefore, it is clear that the chemical relationship between HPLC peak 1 and secondary peak 2 presented in Figure 4 was probably epipyrone A in different ionisation states (e.g., monosalt, disalt, or neutral).

### 2.3. Minimum Inhibitory Concentrations (MICs) of Disalt Epipyrone A

Moulds showed higher resistance against the disalt of epipyrone A compared to yeasts (Table 4). The MIC for filamentous fungi ranged from 0.2 mg.mL^−1^ for *Sclerotinia sclerotiorum* up to 2 mg.mL^−1^ for *Aspergillus oryzae*, whereas the MIC against yeasts varied between 0.03 and 0.04 mg.mL^−1^ (Table 4).

## 3. Discussion

*E. nigrum* is a fungal species that is genotypically and phenotypically variable [2,26], with different strains producing distinct biologically active compounds. The genus *Epicoccum* is taxonomically comprised of a multitude of very diverse fungal species with very unclear phylogenetic markers [26]. This poses a serious obstacle for characterising the secondary metabolite potential of different *Epicoccum* species. Although epipyrone A is a metabolite previously isolated from the culture of a non-identified *Epicoccum* species that presented antiviral activities [3], we have demonstrated here that its disalt is a molecule with broad-spectrum antifungal activity. Due to the complexity of *Epicoccum* taxonomy [26], the lack of full genome sequence for different *Epicoccum* species and the great diversity of secondary metabolites produced by these fungi [3,5,13,14,15,16,17,18,19], it is extremely difficult today to speculate whether or not epipyrone A is produced only by *E. nigrum* or by other *Epicoccum* species as well.

The molecular mass of the two major pigmented compounds (peak 1 and 2, Figure 3) of the methanol extract from *E. nigrum* culture were 612.29. Interestingly, orevactaene/epipyrone A, an inhibitor of HIV-1 Rev protein [21], telomerase [20] and influenza A virus (H1N1) [3], also isolated from *Epicoccum* sp. cultures has the same mass [21]. However, the maximal UV absorption of these purified compounds were obtained at 428 nm, suggesting a slightly different conjugation system compared to orevactaene/epipyrone A, which presents UV max at 432 nm [21].

Burge et al. [13] reported the biological activity of two unidentified pigments produced by a particular strain of *E. nigrum* along with their partial chemical characterisation. Both pigmented compounds appeared to be in a pH-dependent equilibrium with each other. Their chemical properties and UV spectra present striking similarities to orevactaene/epipyrone A [21]. Although their chemical structure(s) were never completely elucidated, it is clear they are also polyenes [13,16]. Subsequently, these compounds were named epirodins [16] and were found to have strong anti-bacterial activity and weak antifungal activity [13,16]. We concluded that epipyrone A disalt could not be any of the two epirodins described by Burge et al. [13] because epirodins were reported to be very unstable and presenting distinct NMR resonances [16]. Besides, we have observed strong antifungal activity and, most importantly, no anti-bacterial activity. Nevertheless, it is clear based on our HPLC analysis of methanol extract obtained from *E. nigrum* cultures that this fungus produces more than one yellow coloured water-soluble metabolite (Figure 4). Therefore, the epicoridins could be other pigment molecules produced by our *E. nigrum* strain that are possibly represented by the less abundant HPLC peaks (Figure 4A) which were not purified and characterised in this study. Nevertheless, *E. nigrum* is reported to be a genetically and phenotypically diverse species [2]. Therefore, it is possible that our particular strain favours the biosynthesis of epipyrone A whist other strains might favour the production of other secondary metabolites, such as epicoridins, flavipins, etc. [13,14], due to various environmental and genetic factors [27,28,29]. Besides, the phenotypic differences between *E. nigrum* strains may give evidence in the future that they are actually different *Epicoccum* species. Certainly, the complexity found in the secondary metabolite literature associated with *Epicoccum* species and their biotechnological importance calls for a taxonomic revision of this group of fungi.

The physicochemical characteristics of the disalt of epipyrone A, on the other hand, is shown to be similar to other polyene antifungal drugs such as nystatin [30]. Interestingly, our results suggest that only the protonated forms of epipyrone A present antifungal activity, since we did not observed any antifungal activity when epipyrone A was assayed at pH below 6. The colour of the compound is yellow–orange, and it showed sensitivity to light in long-term exposure when dissolved in aqueous solution (pH 7), which is a characteristic of polyene compounds [31]. At pH 7 the (mono)-protonated form of epipyrone A was likely to be the major compound in solution. However, this can only be confirmed when the pka values of epipyrone A is determined. Epipyrone A contains a polyene chain, similar in some regards to other polyene-class antifungal drugs, which suggests that its possible mode of action could involve the interaction with ergosterol in the target fungal cell membrane [32]. This polyene/ergosterol interaction results in leakage of intracellular content, causing cell death [33]. However, compared to the other polyene antifungal drugs, the chemical structure of epipyrone A is significantly different as it is linear and not cyclic like other known antifungal polyene drugs such as amphotericin B and nystatin. Besides, epipyrone A possesses α-pyrone moiety that is glycosylated, which is rare in nature, increasing its solubility in water [3,34]. Several antifungal compounds with structural differences to other members of the same antifungal drug family showed significantly altered biological activities [31,34], with varied spectra of antifungal activity [35,36,37], and potentially presented additional modes of action [38,39,40]. Therefore, it is possible that epipyrone A presents a different mode of action compared to other polyene antifungal compounds. Given the current interest in polyene class of antifungal compounds due to their broad spectrum of activity [37,41], further studies on epipyrone A may lead to the development of a new antifungal drug.

The abundant production of epipyrone A by our strain of *E. nigrum* as a secreted compound in culture media makes it an excellent candidate for scale up production of this antifungal metabolite using fermentation technology. The versatile physicochemical properties of epipyrone A also make it compatible with most common delivery strategies of fungicides currently in use in agriculture for example; and its bright yellow colour coupled to its high water solubility (rare among polyene class of antifungal compounds) add a component of easy formulation, traceability and removal if employed as a natural fungicide on crops.

On the other hand, the disalt of epipyrone A also has the potential to be used in the treatment of human fungal diseases. However, its activity still has to be compared to other antifungal drugs currently in use and its feasibility to be used in pharmaceutical products to treat fungal infections in humans and animals will depend of its toxicity. Linked to that is the possibility to use the disalt of epipyrone A as a natural food colorant if it proves to be non-toxic to humans. Being a yellow natural compound that is soluble in water and stable at different temperatures in addition to its antifungal properties, disalt of epipyrone A may be a very attractive natural food colorant or colorant for cosmetic products. Therefore, there are several potential avenues for application of epipyrone A that can be explored.

## 4. Materials and Methods

### 4.1. Microbial Strains

*Epicoccum nigrum* ICMP 19,927 was isolated in 2007 as a likely airborne contaminant of agar plates at the School of Biological Sciences, University of Auckland, and identified based on its full genome sequence (DDBJ/ENA/GenBank under the accession no. NCTX00000000) [42]. It is currently deposited at the International Collection of Microorganisms from Plants (ICMP) maintained by Manaaki Whenua Landcare Research (www.landcareresearch.co.nz/). Other microbial strains used for antimicrobial activity tests are listed in Table 2.

### 4.2. Production, Extraction and Isolation of Antifungal Compounds

*E. nigrum* was cultured on Czapek Yeast Agar (CYA) medium containing: sucrose (30 g.L^−1^), yeast extract (5 g.L^−1^), NaNO_3_ (6 g.L^−1^), K_2_HPO_4_ (1 g.L^−1^), KCl (0.5 g.L^−1^), MgSO_4_·7H_2_O (0.5 g.L^−1^), FeSO_4_·7H_2_O (0.01 g.L^−1^), agar (15 g.L^−1^), pH 6; and on minimal medium (MM) containing: ^13^C_6_-D-glucose (10 g.L^−1^, Cambridge Isotope Laboratories, Inc., Tewksbury, MA, USA), KNO_3_ (5 g.L^−1^), KH_2_PO_4_ (3 g.L^−1^), MgSO_4_·7H_2_O (0.5 g.L^−1^) and vitamins and trace metals as described according to Verduyn et al. [43]. The culture plates were incubated at 25 °C in the dark for 14 days.

Colonized agar from 20 plates of 14 days old *E. nigrum* culture was first macerated into a 1 L glass bottle, by passing it through a 50 mL syringe (Terumo, Japan). The mixture of macerated agar and mycelia was resuspended into approximately 400 mL of analytical grade methanol (−20 °C); then the mixture was shaken at 180 rpm for 1 h, at 4 °C, and protected from light. The fungal biomass and agar debris were removed from the methanol extract through centrifugation at 4000 rpm (Eppendorf^®^ Centrifuge 5804/5804 R), for 30 min, at 4 °C, which was followed by vacuum-aided filtration using first Whatman No. 1 filter paper and then a 0.45 μm membrane filter (Merck Millipore, Darmstadt, Germany). Filtered crude extracts were stored at −80 °C until further experiments.

#### 4.2.1. Solid-Phase Extraction (SPE)

Crude methanol extract of *E. nigrum* was pre-purified using SPE column (Strata SI-1 Silica, 50 μm, 70 A, 10 g/60 mL Gigatubes, Phenomenex, Torrance, CA, USA). The cartridge was primed with 60 mL of Milli-Q H_2_O; then with 60 mL analytical methanol, followed by 60 mL acetonitrile; and finally, with 20 mL of acetonitrile/methanol (1:1 *_v_*_/*v*_) mixture. Approximately 2 mL of crude methanol extract dissolved in acetonitrile/methanol (1:1 *_v_*_/*v*_) mixture were loaded into the cartridge, which was eluted with the same solution mixture. Only the yellow-pigmented fraction was collected and then concentrated to dryness using a solvent evaporator under vacuum (Savant SPS121P Speedvac, Thermo Fisher Scientific, Waltham, MA, USA).

#### 4.2.2. Semi-Preparative Reverse-Phase HPLC

The SPE-purified yellow fraction was dissolved in analytical grade methanol or Milli-Q H_2_O at a concentration of 1 mg.mL^−1^, and then subjected to further purification using a semi-preparative HPLC-DAD system (Shimadzu, Kyoto, Japan) operating at 428 nm, and equipped with a Gemini-NX C18 column (250 mm × 10 mm, 110A, 5 μm, Phenomenex Inc., Torrance, CA, USA). The analytes were separated using solvent gradient using two eluents as mobile phase: A, Milli-Q H_2_O with pH adjusted to 10.0 using ammonium hydroxide; and B, methanol and isopropanol mixture (8:2 *_v_*_/*v*_). The flow rate was set at 3 mL.min^−1^ and column temperature at 25 °C. The gradient was established as follows: from 0–12.5 min the flow of eluent B was increased from 25% to 74%. From 12.5–13.5 min, the flow of B was decreased back to 25% of the flow rate. From 13.5–20 min, the flow of B was kept constant.

#### 4.2.3. Column Chromatography

Larger quantities of antifungal compound were purified from the crude methanol extract using silica gel column chromatography with LiChroprep RP-8 (40–63 μm, Merck Millipore, Darmstadt, Germany). The column was primed with two volumes Milli-Q H_2_O. Then one volume of crude extract diluted with Milli-Q H_2_O (1:5 *_v_*_/*v*_) was loaded into the column. The crude extract was washed with two volumes of Milli-Q H_2_O to remove sugar, and then increments of MeOH in Milli-Q H_2_O were applied to obtain the yellow–orange fraction, which eluted with 40–60%_(*v*/*v*)_ MeOH. The collected fractions were then dried under vacuum using a solvent evaporator (Savant SPS121P Speedvac, Thermo Fisher, Waltham, MA, USA) followed by freeze drying (VirTis freeze-dryer, SP Scientific, PA, USA) to obtain a dark orange powder. These were stored at −20 °C until further experiments.

### 4.3. Structure Elucidation

#### 4.3.1. Ultra-Performance Liquid Chromatography–High-Resolution Mass Spectrometry (UPLC–MS)

The purified yellow fractions were further analysed using an Accela 1250 HPLC (Thermo Fisher Scientific, Waltham, MA, USA) equipped with a Syncronis C18 column (100 × 2.1 mm, 1.7 μm, Thermo Fisher Scientific, Waltham, MA, USA). Analysis of the chromatographic peaks was achieved using a Q-Exactive Orbitrap mass spectrometry system (Thermo Fisher Scientific, Waltham, MA, USA) operating in positive ionization mode. The column and samples were maintained at 30 and 4 °C, respectively. The analytes were separated using ultrapure H_2_O with 10 mM ammonium formate and ammonium hydroxide at pH 9 (eluent A), and acetonitrile with 10 mM ammonium formate and ammonium hydroxide at 3.75%_(*v*/*v*)_ ultrapure H_2_O at pH 9 (eluent B). The column flow rate was kept at 0.4 mL.min^−1^. The gradient was established as follows: From 0–7 min the flow of eluent B was kept constant at 5% of total flow rate. Then, from 7–22 min the flow of eluent B was increased from 5% to 85% of the flow rate, followed by a rapid return to 5 % over 1 min; and it was maintained at that rate for 4 min to finalize the cycle. Centroid MS scans were acquired in the mass range of 50–740 *m*/*z*. The Orbitrap mass spectrometer had a mass resolution of 35,000 (full width at half maximum (FWHM) as defined at *m*/*z* 200), AGC 1 e6, IT 100 ms, sheath gas 50, aux gas 13, sweep gas 3, spray voltage 4.00 kV, capillary temperature 263 °C, s-lens RF 50.0 and heater temperature at 425 °C. Mass calibration was performed prior to each analytical batch using an instrument manufacturer defined calibration mixture.

#### 4.3.2. Nuclear Magnetic Resonance Analysis (NMR)

NMR spectra of the purified sample were recorded with a Bruker Avance AVIII-400 or AVIIIHD-500 spectrometer (Bruker, Karlsruhe, Germany) operating at 400 and 500 MHz for ^1^H nuclei and at 100 and 125 MHz for ^13^C nuclei. Standard Bruker pulse sequences were used. Deuterated methanol (CD_3_OD: δH 3.30, δC 49.00) was used as the internal reference.

#### 4.3.3. Ultraviolet-Visible (UV–Vis)

Spectra were measured with a spectrophotometer model U1800 (Hitachi High-Technologies Corporation, Tokyo, Japan).

### 4.4. Stability Tests

Stability was determined by resuspending between 0.3 to 0.5 mg.mL^−1^ of purified antifungal compound in methanol; in ethanol 5%_(*v*/*v*)_ and water; and in Milli-Q H_2_O at different pHs, and measuring the absorbance at 428 nm every 24 h for 10 days (except for high-temperature stability test). The test samples were kept in glass vials under constant incandescent light at 20 °C (except for the temperature stability test).

### 4.5. Antimicrobial Activity Assays

#### 4.5.1. Agar Diffusion Assays

The bacterial strains were assayed on buffered nutrient agar (NA) plates (peptone 5 g.L^−1^, yeast extract 3 g.L^−1^, NaCl 5 g.L^−1^, agar 15 g.L^−1^, pH 7), and fungal strains were assayed on buffered YPD agar plates (yeast extract 6 g.L^−1^, peptone 3 g.L^−1^, dextrose 10 g.L^−1^, agar 15 g.L^−1^, pH 7.0). Test compound/extract dissolved in phosphate buffer (10 mM, pH 7) were absorbed onto individual paper disks (6 mm dimeter) at 20 µL per disk. Phosphate buffer was used as a negative control. An agar plug containing mycelium (approximately 1 cm in diameter) was inoculated on the centre of agar plates to carry out the assays against filamentous fungi. Suspensions containing 1 to 5 × 10^5^ CFU/mL of each yeast and bacteria were prepared, and 200 µL was spread on each agar surface. The assay plates with filamentous fungi were incubated at 25 °C for 5 days, and the yeast and bacteria assay plates were incubated at 28 °C for 3 days.

#### 4.5.2. Spore Germination Assay

The activity of the antifungal compound against the germination of fungal spores was evaluated using spores aseptically recovered from *Botrytis cinerea* agar plates. Two hundred microlitres of *B. cinerea* spore suspension in sterile YPD broth at pH 7 (10^6^ spores.mL^−1^) was mixed with 1.8 mL of HPLC purified antifungal compound resuspended in sterile YPD broth at pH 6. Three different concentrations of antifungal compound were tested using a 12-well plate—2.7, 1.3 and 0.7 mg.mL^−1^ respectively. The plate was incubated at 25 °C. Each concentration was prepared in triplicate. The germinated spores were observed and recorded at 12, 24, 48, 72, 96, 120, and 240 h. The percentage of spores germinated was determined by microscopic examination of three microscopic fields (haemocytometer) per sample. Spores were considered germinated when the germ tube length was equal or longer than the diameter of the spore. YPD broth without antifungal compound was used as positive control.

#### 4.5.3. Minimal Inhibitory Concentrations (MICs)

MICs of the purified antifungal compound were determined using 96-well plates according to the European Committee on Antimicrobial Susceptibility Testing (EUCAST) method [44] with modifications. The MICs for *S. sclerotiorum* and *A. niger* were determined using spore suspension (10^6^ spores.mL^−1^) in potato dextrose broth (potato extract 4 g.L^−1^, dextrose 20 g.L^−1^) at pH 7 and incubated at 25 °C in the dark under constant agitation at 180 rpm. The MICs for *S. cerevisiae* and *C. albicans* were determined using RPMI1640+MOPS broth [44] at pH 7 and incubated at 28 °C in the dark under constant agitation at 180 rpm with initial inoculation of 5 × 10^5^ CFU/mL. Plates were examined after 24 h, 48 h and 5 days of incubation. The MIC of the compound was defined as the concentration of the purified antifungal compound required to completely inhibit the spore germination of filamentous fungi or the multiplication of yeast cells determined by cell count through an optical microscope.

## Figures and Tables

**Figure 1 molecules-25-05997-f001:**
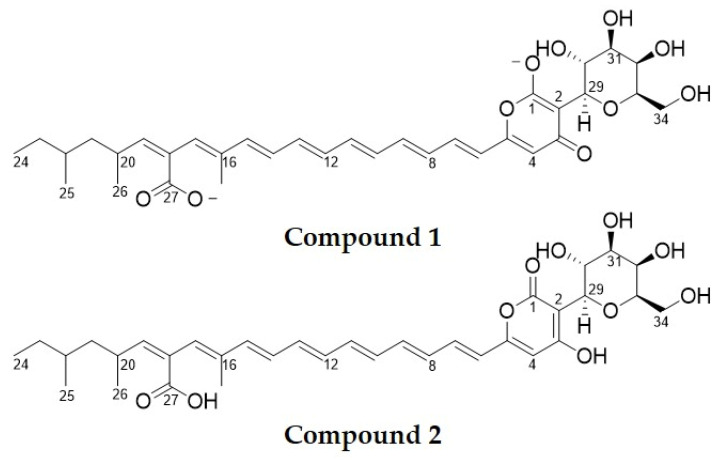
Chemical structure of the purified yellow antifungal compound (compound **1**) and epipyrone A (compound **2**). The purified yellow antifungal compound is a disalt of epipyrone A.

**Figure 2 molecules-25-05997-f002:**
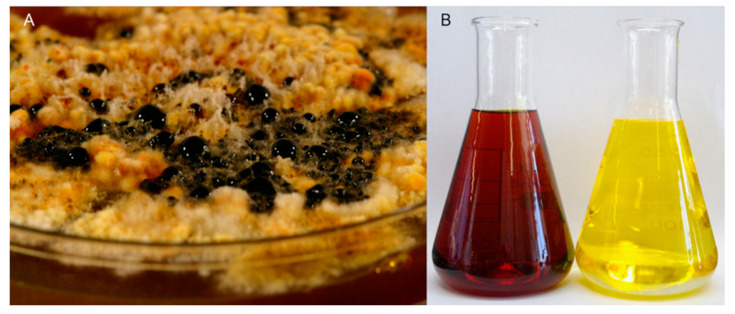
Bioactive exudate produced by *Epicoccum nigrum* ICMP 19927 when growing on complex medium at 25 °C for 14 days (**A**). Crude methanol extract of *E. nigrum* culture in two different dilution (**B**). Diluted crude methanol extract reveals the presence of bright yellow compounds.

**Figure 3 molecules-25-05997-f003:**
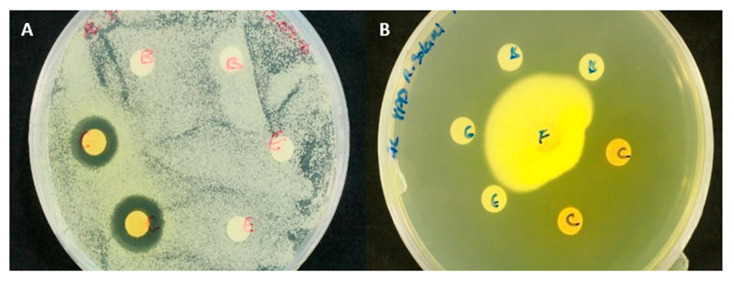
Agar diffusion assays of crude methanol extract obtained from *Epicoccum nigrum* cultures. Different concentrations of crude extract were tested against *Saccharomyces cerevisiae* (**A**) and *Rhizoctonia solani* (**B**). These were compared against solvent blank (paper disks B).

**Figure 4 molecules-25-05997-f004:**
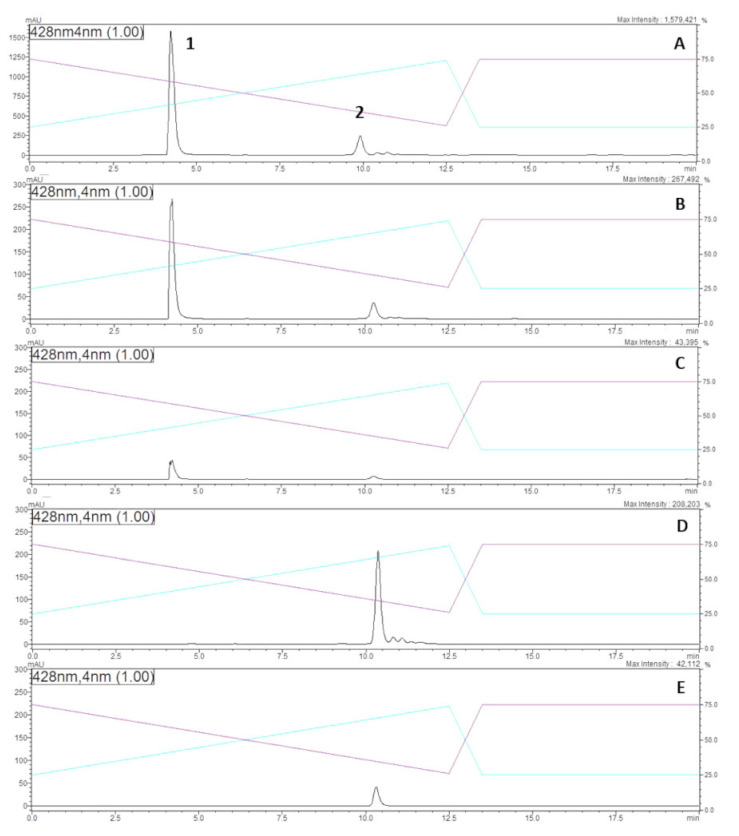
Semi-preparative HPLC-DAD chromatograms of SPE-purified crude methanol extract of *Epicoccum nigrum* recorded at 428 nm using a C18 column. Pink line: eluent A (H_2_O at pH 10); blue line: eluent B (methanol:isopropanol 8:2 *_v_*_/*v*_). **A**: Two major peaks containing yellow compounds were observed at 4 min (peak 1) and 9.5 min (peak 2), in addition to a few minor peaks. **B**: Re-injection of peak 1 (from **A**) dissolved in pure methanol. **C**: Re-injection of peak 2 (from **A**) dissolved in pure methanol. **D**: Re-injection of peak 1 (from **A**) dissolved in H_2_O. **E**: Re-injection of peak 2 (from **A**) dissolved in H_2_O.

**Figure 5 molecules-25-05997-f005:**
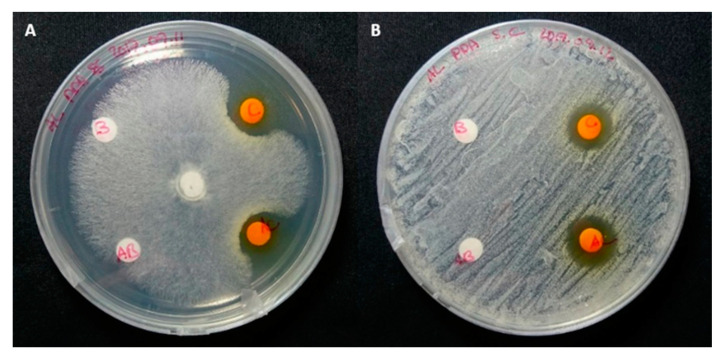
Example of the antifungal activity of a yellow compound purified from the crude methanol extract of *Epicoccum nigrum* ICMP 19927 and dissolved in sterile water. The photograph images show visual results of agar diffusion assay against *Sclerotinia sclerotiorum* (**A**) and *Saccharomyces cerevisiae* (**B**). Inside the plates: white paper discs are negative controls (sterile water) and yellow–orange disks contain purified antifungal compound. All fungal strains listed on Table 1 showed similar results.

**Figure 6 molecules-25-05997-f006:**
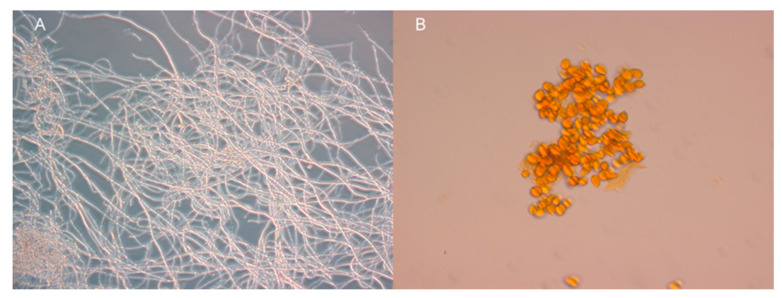
Biological activity test against spore germination of *Botrytis cinera* after 10 days incubation at 25 °C on YPD broth (**A**) and YPD broth containing purified antifungal compound (**B**). Images show light microscopy photographs (×400) of fully germinated mycelia (**A**) and non-germinated spores (**B**).

**Table 1 molecules-25-05997-t001:** Stability of antifungal yellow compound (epipyrone A) in different conditions.

Condition	Specific Condition	Initial Absorbance (428 nm)	Final Absorbance (428 nm) *
pH stability	3	1.46	1.45
	5	1.67	1.67
	7	2.05	2.05
	8	2.09	2.08
	10	2.09	2.09
Stability to light	Milli-Q H_2_O	2.05	1.82
	Methanol	2.04	2.04
	Ethanol 5% *_v_*_/*v*_ in H_2_O	2.04	2.04
Stability to temperature	−20 °C in water	2.08	2.07
	−20 °C in methanol	2.05	2.06
	4 °C in water	2.08	2.06
	4 °C in methanol	2.05	2.03
	60 °C in water (60 min)	2.08	2.06
	60 °C in methanol (60 min)	2.05	2.04
	100 °C in water (60 min)	2.08	2.07
Stability to microwave radiation	Domestic microwave (5 min, 1000 W)	2.05	2.04

* After 10 days (except for hot temperature and microwave radiation tests).

**Table 2 molecules-25-05997-t002:** Antimicrobial activity using disk diffusion assays against epipyrone A disalt.

Species	Strain Designation	Maintenance	* Growth Inhibition
*Alternaria alternata*	ICMP 1099-96	PDA at 25 °C	+
*Aspergillus fumigatus*	SVB-F18	PDA at 28 °C	+
*Aspergillus niger*	ICMP 17511	PDA at 25 °C	+
*Aspergillus oryzae*	ICMP 1281	PDA at 25 °C	+
*Botrytis cinera*	ICMP 16621	PDA at 25 °C	+
*Candida albicans*	MEN	YPD at 28 °C	+
*Enterococcus faecalis*	NCTC 775	NA at 28 °C	−
*Epidermophyton floccosum*	SVB-F16	PDA at 28 °C	+
*Escherichia coli*	W3110	NA at 28 °C	−
*Magnaporthe grisea*	SVF-F2	PDA at 25 °C	+
*Micrococcus luteus*	SVB-B32	NA at 28 °C	−
*Microsporum gypseum*	NZRM2242	PDA at 28 °C	+
*Mucor fresen*	PDD 42019	PDA at 25 °C	+
*Mycosphaerella graminicola*	PDD 12257	PDA at 25 °C	+
*Phomopsis viticola*	ICMP 16419	PDA at 25 °C	+
*Pseudomonas aeruginosa*	SVB-B9	NA at 28 °C	−
*Rhizoctonia solani*	ICMP 11620	PDA at 25 °C	+
*Rhizopus stolonifer*	ICMP 13555	PDA at 25 °C	+
*Saccharomyces cerevisiae*	CEN.PK113.7D	YPD at 28 °C	+
*Sclerotinia sclerotiorum*	ICMP 13844	PDA at 25 °C	+
*Staphylococcus aureus*	ATCC 29213	NA at 28 °C	−
*Trycophyton rubrum*	NZRM3251	PDA at 28 °C	+
*Venturia inaequalis*	PDD 32452	PDA at 25 °C	+

* Presence of halo of growth inhibition observed through disk diffusion assay. **NA:** Peptone (5 g.L^−1^), yeast extract (3 g.L^−1^), NaCl (5 g.L^−1^), agar (15 g.L^−1^), pH 7; **PDA**: Potato (4 g.L^−1^), dextrose (20 g.L^−1^), agar (15 g.L^−1^), pH 7; **YPD**: Yeast extract (6 g.L^−1^), peptone (6 g.L^−1^), dextrose (10 g.L^−1^), agar (15 g.L^−1^), pH 7.

**Table 3 molecules-25-05997-t003:** Comparison of ^1^H and ^13^C NMR data of the yellow antifungal compound **1** purified from *Epicoccum nigrum* ICMP 19,927 culture and epipyrone A (compound **2**) [3] in CD_3_OD.

	Epipyrone A [3]		Compound 1		Compound 2	
No.	δ_C,_ type ^a^	δ_H_ (*J* in Hz) ^b^	δ_C,_ type ^c^	δ_H_ (*J* in Hz) ^d^	δ_C,_ type ^c^	δ_H_ (*J* in Hz) ^d^
1	164.9	-	169.4	-	166.1	-
2	100.4	-	98.9	-	102.1	-
3	169.1	-	181.8	-	170.4	-
4	100.9	6.08, s	110.2	5.81, s	102.3	6.10, s
5	159.4	-	158.4	-	160.7	-
6	121.1	6.18, d (14.8)	124.1	6.11, d (15)	122.4	6.20, d (14.8)
7	136.8	7.12, dd (14.8)	135.0	7.06, dd (15, 10)	138.1	7.14, dd (14.8)
8	130.9	6.43, m	132.4	6.43, m	130.1	6.43, m
9	139.3	6.62, dd	138.6	6.55, dd	140.6	6.64, dd
10	132.7	6.32, m	133.2	6.35, m	134.0	6.32, m
11	132.2	6.41, m	133.1	6.41, m	133.5	6.41, m
12	135.7	6.47, m	137.1	6.45, m	137.0	6.47, m
13	128.8	6.39, m	128.6	6.35, m	132.3	6.39, m
14	136.3	6.34, m	136.7	6.35, m	137.6	6.34, m
15	139.0	6.44, m	141.5	6.35, m	140.3	6.44, m
16	135.3	-	141.1	-	136.6	-
17	130.1	6.13, s	134.3	6.01, s	132.2	6.12, s
18	127.7	-	134.4	-	131.3	-
19	147.8	5.61, d (10.5)	140.4	5.12, d (9.7)	149.0	5.60, d (10.5)
20	31.8	3.03, m	33.1	2.73–2.80, m	33.1	3.02, m
21	44.5	1.12, 1.35, m	46.3	1.32–1.39, m	45.8	1.12, 1.35, m
22	32.5	1.32, m	33.5	1.05, 1.31, m	33.8	1.32, m
23	29.9	1.16, 1.33, m	31.1	1.13, 1.30, m	31.2	1.16, 1.33, m
24	10.4	0.87, t (7.6)	11.6	0.87, t (7.8)	11.6	0.86, t (7.5)
25	18.2	0.85, d (6.0)	20.0	0.87, d (6.8)	19.5	0.85, d (6.0)
26	20.4	1.02, d (6.6)	22.0	1.01, d (6.7)	21.7	1.02, d (6.6)
27	170.7	-	178.3	-	172.0	-
28	12.5	1.88, s	13.3	1.97, s	13.7	1.87, s
29	75.0	4.55, d (9.3)	77.2	4.53, d (9.6)	76.3	4.54, d (9.7)
30	68.9	4.21, t (9.9)	69.4	4.40, dd (9.6)	70.1	4.20, t (9.7)
31	75.2	3.53, dd (2.7, 9.3)	77.5	3.47, dd (9.6, 3.4)	76.5	3.52, dd (2.7, 9.3)
32	69.9	3.93, d (2.7)	71.9	3.85, d (3.4)	71.2	3.93, d (2.7)
33	79.5	3.62, m	80.5	3.58–3.61, m	80.8	3.60, m
34	61.5	3.72–3.76, m	62.8	3.72–3.68, m	62.8	3.72–3.74, m

^a^ 150 MHz for ^13^C NMR [3]; ^b^ 600 MHz for ^1^H NMR [3]; ^c^ 125 MHz for ^13^C NMR; ^d^ 500 MHz for ^1^H NMR.

**Table 4 molecules-25-05997-t004:** Minimum inhibitory concentration (MIC) of disalt epipyrone A purified from *Epicoccum nigrum* ICMP 19,927 against different fungal species.

	Organism	MIC (mg.mL^−1^)
Filamentous fungi	*Aspergillus oryzae*	2.00
	*Sclerotinia sclerotiorum*	0.20
Yeasts	*Saccharomyces cerevisiae*	0.03
	*Candida albicans*	0.04

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
