# Peer review of "Epipyrone A, a Broad-Spectrum Antifungal Compound Produced by *Epicoccum nigrum* ICMP 19927"

_molecules, 2020, doi:10.3390/molecules25245997_

Round 1
Reviewer 1 Report
The manuscript entitled “Epipyrone A, a broad-spectrum antifungal compound produced by Epicoccum nigrum ICMP 19927” by Lee et al., identified and chemically characterized a compound contained in the methanolic extract of E. nigrum. In addition, some tests to evaluate the biological activity of said compound were performed. Some points, which I list below, should be better clarified:
- The structural formula of the compound - Figure 1 must be moved to results and a caption must be added to it.
- Line 51- “CYA plates”: first quote from the acronym- put the meaning and the same in parentheses.
- Line 54 - “This pigmented solution showed a pronounced antifungal activity against other filamentous fungi and yeasts”. Would “pronounced antifungal activity” be shown in Figure 3A?
- Lines 60-68: Figure 3a, b, and e. Was it a mistake? Figure 4?
- Lines 72-76. They must be moved to discussion.
- Line 95- “confirming its broad antifungal activity”. Any compound with antifungal activity must act on both yeast and filamentous fungi.
- The authors list in Table 2, several organisms tested for antifungal activity in the study. However, in Table 4, only the MIC of some are shown, and in Figures 3 and 5, some visual results of the disc-diffusion methodology. I suggest that the authors show in a table all the results (disk-diffusion and MIC) for all the tested fungi.
- The authors detail the chemical characterization of the compound epipyrone A. However, there is also no table with the total chemical composition of the nigrans extract, although the main interest of the authors is clearly the mentioned compound. What is the reason for this specific interest?
- Line 162: “Significant fungicidal activity”. "Significant" implies the use of statistical methods. Apparently they were not used.
- Figure 6 shows the results of the S. cerevisae x epipyrone A. death curve. What is the reason for the choice of the organism among the others in the study?
- Line 205-207. “We concluded that epipyrone A disalt could not be any of the two epicoridins described by Burge et al. [13] because we have observed strong antifungal activity and, most importantly, no anti-bacterial activity”. 1- This sentence must be revised. The authors presented chemical methodologies capable of differentiating isomers (MS fragmentation pattern, photodissociation techniques - UV Vis spectra, absorption radiation wavelength)
- Lines 211- 213: “Given that E. nigrum is genetically and phenotypically very variable [2], it is possible that our particular strain favours the biosynthesis of epipyrone A whist other strains might favour the production of epicoridins, flavipins [13-14]”. Some articles cited below may contribute to improve the authors' incipient discussion on the topic addressed: Calvo et al., 2002 (doi 10.1128/MMBR.66.3.447-459.2002); Yang et al., 2015 (doi 10.1038/ncomms7129); García-Estrada et al., 2018 (doi:10.3390/fermentation4020047)
- Line 257- “Other microbial strains used for antimicrobial activity tests are listed in Table 1.” Was it Table 2? A mistake?
- Line 337- Agar diffusion methodology. The authors described the methodology very succinctly. Extremely important points such as inoculum size, reference drug, reference strains are not mentioned, which can invalidate the results. In addition, the two culture media used (for bacteria and fungi), the incubation temperature and the reading time are not those standardized by EUCAST or CLSI. Would the results be valid? Please see Arendrup et al. (2010), doi 10.1128 / AAC.01256-09. Another question regarding the methodology is what is the cutoff point adopted? What was considered sensitive, intermediate or resistant? A table with the results obtained would also be advisable.
- Line 345 4.5.2. Spore germination assay. What was the reason for the choice of cinnamon Botrytis for these tests? There are no results described for these tests.
- Line 357: In determining the minimum inhibitory concentration using the broth microdilution method, ref. cited - 24, is an additive to the methodology of EUCAST Edef 7.1 (Rodriguez-Tudela et al., 2008, doi: 10.1111 / j.1469-0691.2007.01935.x). The validity of the results is also questionable, since it has no parameters and the performance of the tests did not comply with the international standardization guides. In addition, “The MIC of the compound was defined as the concentration of the purified antifungal compound required to completely inhibit the growth of the tested fungal strains when observed through an optical microscope ”does not mean that the growth of the fungus is inhibited; it may just be a delay (slower). Again, a table with all results must be presented or a justification for showing part of the results (Table 4).
- Line 366 4.5.4. Time-to-kill curve. This assay has several problems. “....80 μL of sample from both flasks were harvested every hour during 24 hours and spread on YPD plates. Three plates were produced per sample at each time point. YPD plates were incubated at 28 ° C for 24 hours ”. If there was a curve during 24h, why was it shown in Figure 6 until 8h? The incubation for 24 hours is not convenient for the growth of filamentous fungus even though it is considered opportunistic (5 days). How the initial inoculum size? What is the quantification limit of the methodology? Why was the concentration tested 4xMIC? Was agitation used? Perhaps a consultation of the literature will help them to reformulate the test: Klepser et al., 1998 (PMID: 9593151); Pfaller et al., 2004 (doi 10.1128 / CMR.17.2.268-280.2004); Appiah et al., 2017 (doi 10.1155 / 2017/4534350).
Reviewer 2 Report
Reviewer’s report Molecules-1003488
The authors have isolated and characterised a disalt od Epipyrone A and charactrised the molecule using a combination of spectroscopic techniques (1D and 2D), and by comparison with the literature data for this compound. Although their spectra were recorded at different frequencies to the literature data, the proton and carbon-13 NMR spectra show striking similarities. The stability of the isolate was evaluated under different pH and thermal (conventional and microwave irradiation) conditions. The compound was evaluated for antimicrobial actvity and found to exhibit significant activity against fungi and less so against bacteria.
Title: Succinct and appropriate
Abstract and Kewords:
The Abstract is brief to the point and well linked to the title and context of the article. Key words are also brief and relevant.
Introduction:
This is well written, free of language or typographical errors, accomapnied by relevant citation.
Results and Discussion
This section biology and chemistry parts is well written and could only pick up the following two minor errors:
- line 89: change 'subject' to 'subjected'
- Line 192: delete 'described'
The authors have also forwaded succinct concluding remarks and projections.
Experimental Section
The methodologies are well written to enable replication by others.
References
Authors should revise the abbreviations of the journal names to be consistent with accepted journal style throughout. Some abbreviations have fullstops and others not. Journal topics be written in a consistent style. refs 4 & 27, for example, has caps for all the keywords while in other references key words in the middle of the sentence are all in small letters. Also some of the abbreviations (eg. Refs 32, 40) need to be in line with approved style. Ref 43 is supposed to be J. Chem. Soc. Perkin Trans. 1 (1 in this case is not volume number).
Recoomendation:
In my view, the results of this study will be of interest to researchers interested in phytochemistry and medicinal chemistry including synthetis. I hereby recommend acceptance of this manuscript for publication in IMolecules pending the inor corrections pointed above and highlighted on the attached PDF to the satisfaction of the editor/s.

Reviewer 3 Report
The manuscript “Epipyrone A, a broad-spectrum antifungal compound produced by Epicoccum nigrum ICMP 19927” [molecules-1003488-peer-review-v1] submitted by Alex J. Lee, Melissa M. Cadelis, Sang H. Kim, Simon Swift, Brent R. Copp, and Silas G. Villas-Boas describes the isolation and structural characterization of a di-anion of epipyrone A. They furthermore check antifungal activity and substance stability at different temperatures and neutral to alkaline pH.
The isolation of epipyrone A from Epicoccum nigrum is not surprising, as there already further studies on this topic have been published. https://doi.org/10.1016/j.bmcl.2020.127242
However, all investigations in the present manuscript are performed with modern and quite common state of the art methods. The overall work seems relatively well planned and the practical works seem to be well performed. The results however, should be checked by some additional experiments and the conclusions should be discussed more critical. In the present form the data and conclusions only slightly further our knowledge about epipyrone A. The manuscript is hence only of limited interest in the fields of Natural Product Chemistry, and to some extent in Pharmaceutical Chemistry.
Therefore, there are some major comments, which should be taken into account by the authors prior to acceptance of the manuscript:
Major Comments:
1) The authors show that they can convert the isolated “compound 1” into epipyrone A by lowering the pH value. They furthermore deduce the structure of the “compound 1” from the spectroscopic data. However, the process of protonation is reversible. The authors are urged to show more precisely to what extent “compound 2” can be converted back into “compound 1” by increasing the pH value. At the same time it would be interesting to determine the two pKa values.
2) The authors should also show that the “compound 1” is not an artifact of the isolation generated by the use of eluents at pH 10 (line 286-287).
3) The authors are encouraged to present the entire MS and NMR spectra (1D and 2D) of the “compound 1” and the resulting “compound 2 / epipyrone A” in a supplementary material.
4) The NMR spectroscopic data of the "compound 2" prepared from "compound 1" should be presented in Table 3. This also enables the reader to compare it with the quoted spectral data from epipyrone A.
5) The authors show that the isolated compound is stable with regard to many influences (temperature, pH value, light, etc.). They use UV absorption for detection. The changes in the absorption at different pH values are interesting, which suggest that different protonated or deprotonated forms of the compound are present. (See also the note on the determination of the pKa value under comment 1). The authors are asked whether they investigated the structure they determined for “compound 1” during the measurements. It is rather obvious that the (mono)-protonated form of epipyrone A was examined in the further measurements at quasi neutral pH value.
6) The same as for comment 4 also applies to all studies of bioactivity. With the pH values present, the deprotonated “compound 1” is probably no longer present, but rather the protonated epipyrone A. Thus, the bioactivities of “compound 1” should be compared in more detail with the bioactivities of epipyrone A described in the literature. Any differences should be discussed against the background of the different protonated or deprotonated forms.
7) In the Materials and Methods part, the authors state DMOS-d6 (sic !, line 324) and CD3OD as deuterated solvents for NMR spectroscopic investigations. However, NMR spectroscopic data are only presented for measurements with CD3OD as solvent. The authors should check whether only one solvent was actually used and whether any differences in chemical shifts could have been influenced by different solvents. Furthermore, the information in Materials and Methods should be corrected.
8) There have been a quite large number of studies on epipyrone A in the last few years. The authors are therefore rasked to determine whether the investigations they have carried out on the deprotonated form of epipyron A really lead to significantly different bioactivities of this form, or whether the data fit into the canon of further reported data on epipyron A. In the latter case, the manuscript should be adapted accordingly.
Round 2
Reviewer 1 Report
The current version of the manuscript shows important changes that provided greater sustainability for the discussion of the authors' meetings.
Reviewer 3 Report
The manuscript “Epipyrone A, a broad-spectrum antifungal compound produced by Epicoccum nigrum ICMP 19927” [molecules-1003488-peer-review-v2] submitted by Alex J. Lee, Melissa M. Cadelis, Sang H. Kim, Simon Swift, Brent R. Copp, and Silas G. Villas-Boas has been revised by the authors. The reviewer is grateful to the authors for taking to heart the previous comments and for addressing several concerns.
All comments have been addressed in the manuscript or in the answer to the reviewers comments. The changes improve the equality and readability of the manuscript. The reviewer is still of the opinion that determining the pKa values would be very helpful. However, the authors state that the determination can currently not be carried out with the quantities of the natural products available.
The manuscript hence now fully and intelligibly describes the isolation and structural characterization of a di-anion of epipyrone A on the basis of the available data. The manuscript should hence be acceptable for publication in Molecules.